# The Polish adaptation of the work role performance questionnaire

Jaroslaw Grobelny[1]*, Mateusz Paliga[2], Olga Zwardon-Kuchciak[3], Mateusz Hauk[3]

**1** Faculty of Psychology and Cognitive Science, Adam Mickiewicz University, Poznań, Poland, **2** Institute of Psychology, Faculty of Social Sciences, University of Silesia in Katowice, Katowice, Poland, **3** Faculty of Educational Sciences, University of Lodz, Łódź, Poland

☯ These authors contributed equally to this work.
* jaroslaw.grobelny@amu.edu.pl

## Abstract

The cultural transferability and relevance of work performance constructs across various cultures remain crucial for global human resources research. This paper investigates the cultural adaptation of the Work Role Performance Questionnaire (WRPQ) for the Polish work environment. Originating from the work role theory, WRPQ encapsulates nine distinct work roles derived from two dimensions: behavior type (proficiency, adaptivity, proactivity) and exhibition level (individual, team, organization). We proposed two hypothesis series addressing the questionnaire's structure and differential predictions by various work-related traits and attitudes. A multidisciplinary team proficiently translated the questionnaire, which was then tested in a pilot study (n = 276) followed by two primary studies involving supervisors (n = 698) and employees (n = 534). The analytical approach integrated confirmatory factor analysis, path analysis, and psychometric evaluations. The findings reinforce the questionnaire's reliability within the Polish cultural context, validate its nine subdimension structure, and elucidate distinct relations with predictors. The successful cultural adaptation of the WRPQ showcases the potential for broader applications in diverse cultural settings, underscoring the importance of context-specific measurement tools in work performance and human resources research.

## Introduction

The efficient measurement of work performance is pivotal in understanding the dynamics of employee productivity. As workplaces become increasingly diverse and globalized, the need emerges to ensure that assessment tools maintain their validity and reliability when applied to various cultural contexts. The development of culturally appropriate tools to measure job performance is essential for fostering a more inclusive and globally representative scientific inquiry within the field of human resource management, ensuring that evaluations accurately reflect diverse workforce contexts

**Data availability statement:** The information needed to reproduce all of the reported results is available at https://doi.org/10.17605/OSF.IO/XB3TC

**Funding:** The author(s) received no specific funding for this work.

**Competing interests:** The authors have declared that no competing interests exist.

and contribute to international research rigor. To address this concern, the present paper endeavors to adapt the Work Role Performance Questionnaire (WRPQ; [1]) to Polish culture. By investigating the relevance of the questionnaire within a given setting, this paper aims to enhance cross-cultural understanding and facilitate a valid evaluation of work performance.

In this research, we conducted a series of studies to evaluate the reliability and validity of the adopted WPRQ as a measure of job performance grounded in role theory. The process began with a pilot study (n = 276), followed by two main studies: one involving supervisors (n = 698) and another focused on employees (n = 534). Across all studies, the WPRQ consistently demonstrated robust psychometric properties, confirming its suitability for both rigorous scientific research and practical applications in workplace settings. The study makes several contributions to existing literature. Foremost, it validates work role performance theory within a new cultural context, which broadens the current comprehension of the cultural adaptability of the work performance construct. Second, the research represents one of the first substantive adaptations of the discussed questionnaires. Through these contributions, the study enhances the reliability and validity of future research findings in the field of human resource management, specifically within a cross-cultural context.

### Work role performance theory

The understanding of work performance has evolved to cover a broad context due to increased workplace complexity. The concept of Griffin et al. [1] accurately captures this evolved perspective, which posits that factors of the work context, such as uncertainty and interdependence, dictate the behaviors valued by organizations. Uncertainty influences the degree to which work roles can be formally defined, which determines whether an individual can excel through compliance alone or if adaptability is necessary. Interdependence establishes how closely work roles are integrated into the broad social structure of an organization (i.e., whether individual performance hinges solely on managing their role or involves contributing to the wider social context). Role theory underscores that work roles are intrinsically tied to organizational contexts [2]. However, previous uses of role theory have prioritized role development instead of performance structure [3, 4]. Griffin et al. addressed this gap by presenting a model that merges two principal characteristics, namely, the behavior types that an employee can manifest within their roles and the levels at which these behaviors resonate with the organizational context.

The type of behavior distinguishes between formal and emergent roles, which identifies three components, namely, proficiency (adherence to clear role standards), adaptability (adjustment to changes in work roles), and proactivity (self-initiated action to anticipate or prompt changes; [1]). The second attribute is role level (individual, team, or organizational) and underscores the importance of interdependence within organizations, which highlights behaviors that foster the social context beyond individual tasks. Role theory posits that individual behaviors in organizations affect personal and collective outcomes based on their fit with the social and role interdependence contexts [5].

Ultimately, the work role performance model of Griffin et al. [1] intersects these two attributes (each classified across three levels), which results in nine distinct work roles. *Individual task proficiency* (ITP) represents distinct behaviors that measure the alignment of employees with the expectations of their roles, independent of social context. It closely aligns with concepts, such as task or in-role performance or job-specific proficiency [6,4]. *Team member proficiency* (TMP) denotes formalized behaviors integrated into a group context. It reflects the degree to which an individual fulfills the expectations of their role as a team member. *Organization member proficiency* (OMP) refers to behaviors that can be formalized and are part of the organizational context; they indicate the extent to which an individual meets the expectations of their role within the organization. OMP shares similarities with contextual performance, organizational loyalty, and organization role behaviors [7,8]. *Individual task adaptivity* (ITA) is the ability to respond and adapt to unexpected changes in work requirements due to dynamic and unpredictable factors. *Team member adaptivity* (ITA) reflects how effectively individuals handle, adapt to, and assist with changes that impact their roles within the team, which emerges from external environments or internal team dynamics. *Organization member adaptivity* (OMA) indicates how individuals adapt to changes that impact their organizational roles. *Individual task proactivity* (ITV) entails taking the initiative to proactively adopt forward-thinking measures, which enhances the roles, work conditions, or personal development of employees. *Team member proactivity* (TMV) relates to the extent to which an individual initiates future-oriented actions to alter the situation or functioning of the team. *Organization member proactivity* (OMV) relates to individual willingness to take self-initiated, forward-thinking actions to change the organization. These proactive behaviors surpass the job responsibilities of employees and contribute to the goals of the organization.

Griffin et al [1] introduced the WRPQ, which intends to examine the nine dimensions of work role. It links the roles and certain traits and attitudes of each work, such as role clarity, openness, role breadth self-efficacy, team support, and organizational commitment, and explores proactivity and suggestion making. The study affirmed the reliability of the WRPQ in assessing work performance and offered a streamlined framework for navigating the complexity of work performance.

## Cross-cultural issues with work performance

Although the research on job performance has been extensive and prolific, the role of cross-cultural differences in performance ratings receives scant attention [9,10]. This is surprising, given the importance of appraisal methods and systems for managing a global and diverse workforce. DeVoe and Iyengar [11] note that the accuracy of performance ratings must be preceded by the comprehension of the role of cultural differences in assessment. Without this understanding, differences in work-related values may lead to low levels of applicability and acceptance of appraisal methods developed in other countries [12].

The abovementioned issues may flaw the use of the WRPQ in the case that no national adaptations exist. Nevertheless, the method has been used in many variations, including the utilization of either all or only selected items [13,14,15,16,17], the calculation of all nine [18,19] or merely a few dimensions [20], and at times, application inconsistent with the underlying theory [16]. In all cases, except for Silva de Carvalho Chinelato [15], the researchers only verified internal consistency. To the best of our knowledge, the only test of the construct validity of the WRPQ was conducted by Saviera et al. [21] in Indonesia, who confirmed the second-order factor model with proficiency, adaptivity, and proactivity as the subdimensions of overall performance. A notable absence of efforts toward rigorous cultural adaptation is observed in prior applications of the WRPQ. These practices pose challenges in synthesizing the existing knowledge of work performance across cultural contexts. The issues highlighted in the review, the need for a systematic adaptation effort becomes increasingly evident and critical for advancing the field.

## Factor structure of subdimensions

The current study design is closely aligned with the original questionnaire process to validate the adequacy of the original and Polish versions. Moreover, as the field has evolved, and certain limitations of the original study have been identified,

a number of hypotheses have been refined or updated in specific areas. First, the primary objective was to validate the ability of the adopted questionnaire to identify the nine work roles, which are defined as factors that would load the questionnaire results. Consistent with the original research, incorporating ratings from supervisors and employees was essential. Supervisor ratings are the gold standard for evaluating job performance, but they remain susceptible to bias [6,22,23]. Conversely, self-ratings are generally more lenient but are based on a comprehensive understanding of ' 'one's work [24]. Therefore, the study deemed that confirming the structure of the results within both samples is critical for establishing the validity of the adopted questionnaire.

*H1a. Supervisors systematically differentiate work behaviors in terms of the nine work roles when rating employees using behavioral items included in the adapted questionnaire.*

*H1b. Employees systematically differentiate work behaviors in terms of the nine work roles when self-rating behavioral items included in the adapted questionnaire.*

## Differential predictors

In line with role theory, outcomes within a role are dependent on its salience and on the specific elements that contribute to effectiveness in such a role. Consequently, the WRPQ results should facilitate the identification of the nine work roles and demonstrate differential validity. Thus, the results are hypothesized to exhibit stronger associations with predictors that are theoretically more significant for the selected role compared with the others.

ITP is closely associated with in-role performance [1]. Consequently, role clarity—the level of clarity possessed by individuals in relation to the expectations and requirements of their roles—is expected to predict these formally mandated behaviors [25,5]. A strong correlation with ITP is expected, because formal requirements primarily apply to individual responsibilities.

Consequently, the study predicts that conscientiousness will exhibit a more pronounced association with ITP than do the other roles. The greater the awareness of the formalized requirements of a specific role, the more conscientiousness is expected to contribute to the effective execution of the role. Furthermore, this trait has consistently demonstrated a robust relationship with task performance [26], which aligns with ITP.

*H2a. (a) Conscientiousness and (b) role clarity will show significantly stronger standardized path coefficients in predicting ITP compared to other work roles.*

Next, the study identified a construct associated with proficiency roles. Achieving success hinges on the determination to pursue goals and the resilience to overcome obstacles, which are qualities that contribute to self-efficacy [27]. Individuals with strong self-efficacy beliefs approach tasks with confidence, are inclined to set ambitious goals, and invest effort to achieve them. Conversely, the exact requirements for adaptivity and proactivity cannot be specified; as such, the discussed predictor is not anticipated to exhibit a strong association with these sets of roles.

*H2b. Work-related self-efficacy will show significantly stronger standardized path coefficients in predicting roles involving proficiency (ITP, TMP, and OMP) compared to other work roles.*

We subsequently identified the predictor of adaptivity roles. Proactivity and adaptivity become more relevant when role requirements cannot be formalized due to uncertainty [1]. In uncertainty, moreover, certain aspects of a role must dynamically emerge through interactions between the role incumbent and the environment, which results in changes. Individuals more receptive to change are predicted to exhibit a more positive response and flexibility in the face of such demands

[28]. Conversely, the openness trait is indirectly connected to the initiation of these changes, such that a notable correlation between this trait and proactivity-related roles was not expected.

*H2c. Openness to change will show significantly stronger standardized path coefficients in predicting roles involving adaptivity (ITA, TMA, and OMA) compared to other work roles.*

For proactivity, the study selected a personality trait associated with the independent initiation of actions and changes as a differential predictor for these roles [29]. We drew from proactivity theory, which describes behaviors under conditions such as uncertainty, unpredictability, and momentariness [30]. Therefore, we anticipated that individuals with the trait associated with the intentional initiation of immediate actions would also exhibit such behaviors in the context of their work roles.

*H2d. The personality trait proactivity will show significantly stronger standardized path coefficients in predicting roles involving proactivity (ITV, TMV, and OMV) compared to other work roles.*

The preceding hypotheses pertain to the types of behavior. Therefore, we selected predictors related to personality and individual characteristics. The model of Griffin et al. [1] also introduces the level dimension. Hence, the study expected that attitudes and organizational-level factors could predict the extent to which behaviors manifest at the individual, team, or organizational level.

Initially, the study proposed that the work organization would facilitate roles at the individual level. This characteristic encompasses the structure and the control granted in task execution [31]. The appropriateness of assigning tasks to match the competence of employees hinges on the work organization. Hence, it should support individual roles by enhancing the clarity of role expectations and allocating requisite resources accordingly.

*H2e. Work organization will show significantly stronger standardized path coefficients in predicting individual task roles (ITP, ITA, and ITV) compared to other work roles.*

The chance for individuals to contribute at the team or organizational level is contingent on the social embeddedness of their roles [5]. Role and identity theories clarify that the greater the sense of belonging and identity recognition, the more inclined individuals will be to support the objectives of the group [4]. Hence, when individuals perceive their teammates as supportive, they respond by actively contributing to the comprehensive success of their team.

*H2f. Perceived team support will show significantly stronger standardized path coefficients in predicting team member roles (TMP, TMA, and TMV) compared to other work roles.*

Griffin et al. [1] examined organizational commitment as a predictor of organizational-level roles. However, commitment is viewed more as a psychological state and less than an organizational-level characteristic [32]. Therefore, we opted to investigate perceived organizational support—an evaluation of the extent to which an employee believes that the organization cares about their well-being in exchange for recognizing their contributions [33]. According to the social exchange model, the relationship between an individual and an organization is transactional in nature [34]. Consequently, the more support an individual perceives, the more they engage in organization-related roles and display behaviors that benefit the company in various ways.

*H2g. Perceived organizational support is more strongly related to organization member roles (OMP, OMA, and OMV) than do other work roles.*

## Methodology

### Development and translation process

The study conducted questionnaire adaptation on the basis of the guidelines of the International Test Commission [35]. As such, the permission of the original authors was obtained, and a theoretical and empirically-based discussion of whether the construct of interest is relevant for a target population and adaptation was carried out.

Next, a multidisciplinary team of experts in work performance and test construction was assembled, including a professional translator fluent in English and Polish with expertise in psychology, along with a scholar specialized in the Polish language. We used a mixed-method approach to prepare the translated questionnaire. First, two work psychologists conducted independent forward translations of the WRPQ. These versions were harmonized during a workshop that included another team member who was excluded from the previous stages. The consensus draft was subsequently submitted to a professional translator tasked with conducting a back-translation. Next, we compared the original questionnaire with the back-translated version and collaboratively resolved inconsistencies. Finally, we sent the subsequent draft to an expert in the Polish language along with a theory-driven explanation of each item. The objective of the expert was to evaluate whether these intentions were accurately conveyed. Following a review by the authors of the proofread version, consensus was reached on language adjustments. Subsequently, the finalized version (available via Open Science Framework at https://osf.io/8uq7s) was advanced to a pilot study.

### Sample and sampling process

A total of 276 individuals participated in the pilot study. The inclusion criteria were being active in the workforce, functioning within team environments, and occupying nonmanagerial positions. Out of the total, 256 met the criteria. The sample predominantly comprised women (79.68%) with an average age of 35.6 years ($SD$ = 9.02). The participants held 12.85 years of professional experience ($SD$ = 8.91). Regarding occupational roles, 71.1%, 1.2%, and 18.8% of the participants were engaged in office-based work, manual labor, and both, respectively. The study employed convenience sampling due to the exploratory nature of the pilot. The participants were recruited through social media groups linked to major Polish cities from April 15th to 16th, 2023. The intended sample size was set to 100, with a stopping rule to halt recruitment at the end of the day of reaching that target. The sample size was determined based on the minimum assumed ratio of subjects per number of items [36,37].

Study 1 engaged a group of 750 individuals. The inclusion criteria stipulated current employment and team management responsibilities, and 698 people were eligible. The sample featured a balanced gender distribution with women comprising 51%. The average age was 41.33 years ($SD$ = 1.11); 7.7% and 48.9% held less than five years and more than 15 years of experience, respectively. Managerial responsibilities included overseeing teams with an average size of 16 and spanned office-based roles (63.9%), manual labor (16%), or a combination of both (20%). The participants were randomly selected from a research panel database from April 19th to 21st, 2023, and nonmonetary gifts were provided to deter automated responses. Sample size calculation for CFA was employed to estimate the intended number of participants. Predicting a moderate effect size (.25), a statistical power of.95, a significance level of.05, nine latent variables, and 27 observables, the study determined the required sample size at 403 [38]. To accommodate potential data loss and adhere to the *very good* sample size recommendation by Comrey and Lee [9], the study aimed to recruit approximately 500 participants. A stopping rule was also used, that is, concluding the data collection upon reaching the intended sample size at the close of the day.

Study 2 engaged 635 participants. The inclusion criteria were being currently employed, holding nonmanagerial positions, and working as team members. The sample of 534 people who met these criteria comprised 68.3% women. The average age reached 41.6 years ($SD$ = 1.75); 12.4% and 51.9% held less than five years and more than 15 years of experience, respectively. Job roles were diversified with 63.9%, 28.1%, and 8.1% engaged in office work, manual labor, and

both, respectively. The sampling approach and rationale for the sample size were consistent with Study 1, with the participants' recruitment carried out from April 19th to 20th, 2023.

## Measurements

### Work role performance

The study used the proposed Polish version of WRPQ, which comprises 27 items, to assess work role performance and evaluate the nine roles (subdimensions). Each role was gauged using three items that elucidated the behaviors that individuals may have displayed in the preceding month. Items were rated using a six-point Likert-type scale ranging from *never* to *very often*. It was determined [1] that all subdimensions and the overall results exhibited a high level of reliability, which exceeded $\alpha = .83$.

### Performance predictors

Conscientiousness was assessed using the relevant scale from the International Personality Item Pool—Big Five Markers-20 questionnaire (IPIP-BFM-20; [39]). This scale was formulated on the basis of the International Personality Item Pool (IPIP; [40]) and validated and adapted for the Polish context. The scale comprises four items that each describe a specific characteristic of an individual. Items were rated using a seven-point Likert-type scale to evaluate how well the descriptions matched the participants.

Role clarity was assessed using the relevant scale from the Polish version of the Copenhagen Psychosocial Questionnaire (COPSOQ; [41,42]). This scale comprises three questions that explore the comprehension of the participants about their roles. Items were rated using a seven-point Likert-type scale ranging from *to a minimal extent* to *to a very high extent*.

To measure openness to change, the study used the openness scale with four items from the IPIP-BFM-20 [39]. The scale demonstrated satisfactory reliability and was established and scored in a manner consistent with that of conscientiousness.

The study then assessed proactivity using the General Proactivity Scale (SPO) of Bańka [29], which is designed to gauge the predisposition toward proactive behavior as a form of creative adaptation for the future. Five items were selected from the subscale of the General Proactivity Scale. Items were rated using a seven-point Likert-type scale.

Work organization was evaluated using a subscale from the Polish adaptation of the Organizational Climate Questionnaire by Rosenstiel and Boegel (OCQ; [31]). This subscale features five items, which were rated using a seven-point Likert-type scale.

Perceived team support was assessed using three questions from the Social Support From Colleagues subscale of the COPSOQ. Items were rated using a five-point Likert-type scale ranging from *never or almost never* to *always*.

Perceived organizational support was subsequently examined using five items from the Polish version of the Perceived Organizational Support Scale (PSWO; [33,34]). All items were rated using a seven-point Likert-type scale.

### Study procedure

After developing the questionnaire, the research project was executed in three phases, namely, a pilot study and two primary studies. An observational pilot study was conducted to initially evaluate the psychometric properties of the adopted WRPQ. The participants completed an online form after providing informed consent. The informed consent was collected in writing via an online form. The criteria for advancing the questionnaire to further studies were as follows: no subdimension should reach "Cronbach's alpha or "McDonald's omega coefficients less than .60, and the mean coefficients for all subdimensions should be no less than .75.

Ultimately, two primary studies were conducted, which were observational and employed a quota sampling executed by a research agency. The participants completed the online form after providing informed consent. In both cases, the informed consent was collected in writing via an online form. These studies aimed to replicate the approach taken by the authors of the original questionnaire [1], who utilized two types of samples, namely, managerial and nonmanagerial, to assess the properties of their tool.

In Study 1, managers were asked to complete the WRPQ twice, as if they were evaluating two of their employees—one with high and another with poor or mediocre performance. A total of 1,318 participants were evaluated in this manner, representing the full sample size used for further analyses in Study 1. Demographic data were collected as control variables. In Study 2, workers assessed their behaviors using the WRPQ and filled out additional questionnaires to measure performance predictors and outcomes. In line with publication ethics, informed consent was obtained from all participants prior to their involvement in the study via online form. Participants were provided with detailed information about the study's objectives, procedures, and their rights, ensuring an informed decision to participate. The research protocol was reviewed and approved by the Research Ethics Committee of the University of Łódź (decision no. 1/KEBN-UŁ/IV/2022–23).

### Analytical strategy

Data were analyzed using the R language (version 4.1.2; [43]) and JASP (version 0.16.0). The following packages were employed: *psych* for reliability testing and factor analysis, and *lavaan* for CFA [44,45].

We performed descriptive statistics, reliability, and preliminary factor analysis at the initial stage of data analysis. Factor loadings for corresponding subdimensions were calculated for each item. Based on the findings of the original study and recommendations in the literature [9], factor loadings should be no less than.60. Given the improved sampling and large sample sizes in the primary studies, the criteria for the alpha and omega reliability coefficients were set to more than.70 and.80, respectively.

To test Hypothesis 1, the study separately conducted CFA for each sample. Six models were compared, namely, (a) a single-factor model, (b) a two-factor model (task and context performance), (c) a three-factor model (task, team, and organization), (d) another three-factor model (proficiency, adaptivity, and proactivity), (e) a six-factor model, and (f) a nine-factor model. These models were compared using various fit and error measures, including $\chi^2$, GFI, AGFI, TLI, PGFI, RMSEA, and SRMR. These indices were selected to encompass a range of fit measures, including absolute, relative, and parsimonious fit as well as noncentrality and absolute residual-based indices. To confirm the hypothesis, the nine-factor model must be superior. Multi-group CFA supplemented the analysis to demonstrate measurement invariance [46].

To test Hypothesis 2, we conducted path analysis using the methodology of the original study [1]. The path coefficients of each of the nine Polish WRPQ subdimensions were compared to each of the eight hypothetical predictors. Consistent with the original paper, for Hypothesis 2 to be confirmed, the path coefficient for the relationship specified in the hypothesis needs to be superior to the other coefficients for a given role.

To ensure the quality of the data, two attention checks were included per survey. Additionally, after each survey, the participants were asked a direct question regarding the reliability of their responses for scientific use.

## Results

### Pilot study

Tables 1 (item-level statistics with factor loadings) and 2 (scale-level statistics) delineated the results of the pilot study. Items pertaining to individual-level roles and proficiency-oriented behaviors displayed substantial kurtosis. However, at the scale level, only ITP exceeded the acceptable threshold for kurtosis [47]. Along with moderate to high skewness, this result hints at potential leniency and self-presentation biases when self-rating this particular role. Notably, the general result exhibited minimal kurtosis and skewness.

**Table 1. Item-level descriptive statistics and exploratory factor loadings, pilot study.**

| Item | | Descriptive statistics | | | | | Factor | | | | | | | | |
|------|------|------------------------|-----|------|------|------|-----|-----|-----|-----|-----|-----|-----|-----|-----|
| | | M[95% CI LL, UL] | SD | Kurt | Skew | W | 1. | 2. | 3. | 4. | 5. | 6. | 7. | 8. | 9. |
| 1 | ITP 01 | 5.67 [5.6,5.73] | .53 | 5.62 | −1.59 | .61*** | .75 | | | | | | | | |
| 2 | ITP 02 | 5.54 [5.45,5.62] | .66 | 5.33 | −1.77 | .69*** | .70 | | | | | | | | |
| 3 | ITP 03 | 5.46 [5.38,5.55] | .69 | .16 | −.76 | .73*** | .44 | | | | | | | | |
| 4 | ITA 01 | 5.25 [5.15,5.34] | .78 | 2.08 | −1.46 | .75*** | .36 | .63 | | | | | | | |
| 5 | ITA 02 | 5.16 [5.05,5.27] | .89 | .51 | −1.17 | .74*** | | .54 | | | | | | | |
| 6 | ITA 03 | 4.55 [4.41,4.70] | 1.21 | .39 | −1.04 | .89*** | | .86 | | | | | | | |
| 7 | ITV 01 | 4.61 [4.48,4.74] | 1.06 | .15 | −.50 | .89*** | | | .79 | | | | | | |
| 8 | ITV 02 | 4.53 [4.38,4.67] | 1.19 | −.02 | −.65 | .89*** | | | .60 | | | | | | |
| 9 | ITV 03 | 4.20 [4.05,4.35] | 1.25 | −.25 | −.44 | .92*** | | | .65 | | | | | | |
| 10 | TMP 01 | 4.79 [4.65,4.93] | 1.16 | .29 | −.85 | .86*** | | | | .53 | | | | | |
| 11 | TMP 02 | 5.34 [5.23,5.44] | .84 | 2.20 | −1.41 | .74*** | | | | .88 | | | | | |
| 12 | TMP 03 | 5.52 [5.44,5.61] | .71 | 3.30 | −1.67 | .67*** | | | | .44 | | | | | |
| 13 | TMA 01 | 5.00 [4.87,5.13] | 1.06 | 2.64 | −1.43 | .8*** | | | | | .50 | | | | |
| 14 | TMA 02 | 4.65 [4.50,4.79] | 1.18 | .20 | −.71 | .88*** | | | | | .74 | | | | |
| 15 | TMA 03 | 4.53 [4.39,4.67] | 1.15 | .59 | −.78 | .89*** | | | | | .81 | | | | |
| 16 | TMV 01 | 4.15 [3.99,4.31] | 1.30 | .11 | −.65 | .9*** | | | | | | .85 | | | |
| 17 | TMV 02 | 3.85 [3.68,4.03] | 1.42 | −.56 | −.36 | .92*** | | | | | | .95 | | | |
| 18 | TMV 03 | 3.88 [3.71,4.05] | 1.37 | −.60 | −.40 | .92*** | | | | | | .90 | | | |
| 19 | OMP 01 | 4.43 [4.25,4.60] | 1.41 | −.30 | −.66 | .88*** | | | | | | | .78 | | |
| 20 | OMP 02 | 3.82 [3.63,4.01] | 1.56 | −.90 | −.30 | .92*** | | | | | | | .84 | | |
| 21 | OMP 03 | 4.49 [4.33,4.65] | 1.30 | −.23 | −.66 | .89*** | | | | | | | .85 | | |
| 22 | OMA 01 | 4.30 [4.13,4.46] | 1.35 | .42 | −.94 | .87*** | | | | | | | | .88 | |
| 23 | OMA 02 | 4.56 [4.42,4.71] | 1.16 | 1.46 | −1.08 | .85*** | | | | | | | | .78 | |
| 24 | OMA 03 | 4.32 [4.16,4.49] | 1.32 | .30 | −.81 | .89*** | | | | | | | | .42 | |
| 25 | OMV 01 | 3.59 [3.40,3.78] | 1.55 | −.97 | −.22 | .92*** | | | | | | | | | .84 |
| 26 | OMV 02 | 3.68 [3.49,3.87] | 1.56 | −.97 | −.33 | .91*** | | | | | | | | | .94 |
| 27 | OMV 03 | 3.16 [2.97,3.35] | 1.51 | −1.02 | .03 | .92*** | | | | | | | | | .78 |

*Note*. CI—confidence interval; LL—lower limit; UL—upper limit; W – Shapiro-Wilk's test for normality of distribution.

***p <.001

Given the cross-cultural adaptation context, an exploratory factor analysis (EFA) was conducted prior to CFA to examine the underlying structure without presuming invariance. The sampling adequacy indices supported the suitability of the data for factor analysis (KMO overall MSA =.88; Bartlett's test of sphericity: $\chi^2$(26) = 947.05, $p$ <.001). All item loadings exceeded the significance threshold of.35 recommended for the obtained sample size [47]. Only one item showed significant cross-loading on two factors; however, as the difference in loadings was greater than.20 and both items belonged to the same role level (individual), this was not considered problematic for subsequent CFA. The factor solution presented in Table 1 shows that items loaded clearly on their respective latent constructs, providing initial support for the proposed measurement structure.

Regarding reliability, all the coefficients of the subdimensions surpassed the set criteria, with the general score being exceptionally reliable (.92 for alpha and omega). Additionally, average variance extracted (AVE) within each subdimension surpassed the correlations with other subdimensions, offering preliminary evidence of differential validity. Taken together, these results supported the endorsement of the initial questionnaire for further assessment.

**Table 2. Scale-level descriptive statistics, pilot study.**

| Scale | | Descriptive statistics | | | | | Reliability | | Correlation coefficients | | | | | | | | | |
|---|---|---|---|---|---|---|---|---|---|---|---|---|---|---|---|---|---|---|
| | | M [95% CI LL, UL] | SD | Kurt | Skew | W | α | ω | 1 | 2 | 3 | 4 | 5 | 6 | 7 | 8 | 9 | 10 |
| 1 | General | 4.56 [4.47,4.64] | .68 | −.15 | −.83 | .99* | .92 | .92 | – | | | | | | | | | |
| 2 | ITP | 5.56 [5.50,5.62] | .48 | 3.91 | −1.27 | .84*** | .65 | .66 | .45 | (.65) | | | | | | | | |
| 3 | ITA | 4.99 [4.89,5.08] | .76 | .32 | −.55 | .90*** | .66 | .67 | .67 | .43 | (.71) | | | | | | | |
| 4 | ITV | 4.45 [4.32,4.57] | 1.01 | 1.91 | −1.18 | .96*** | .83 | .85 | .73 | .31 | .56 | (.80) | | | | | | |
| 5 | TMP | 5.22 [5.13,5.30] | .68 | 1.97 | −1.04 | .89*** | .60 | .63 | .43 | .35 | .30 | .20 | (.64) | | | | | |
| 6 | TMA | 4.73 [4.61,4.84] | .95 | −.18 | −.50 | .92*** | .80 | .80 | .78 | .26 | .55 | .54 | .30 | (.75) | | | | |
| 7 | TMV | 3.96 [3.80,4.12] | 1.27 | .56 | −.41 | .96*** | .93 | .93 | .70 | .18 | .25 | .54 | .23 | .48 | (.90) | | | |
| 8 | OMP | 4.25 [4.09,4.40] | 1.28 | 1.22 | −.98 | .95*** | .88 | .88 | .61 | .26 | .33 | .32 | .23 | .40 | .17 | (.84) | | |
| 9 | OMA | 4.39 [4.26,4.53] | 1.12 | −.84 | −.23 | .92*** | .85 | .85 | .74 | .21 | .51 | .42 | .24 | .63 | .39 | .39 | (.82) | |
| 10 | OMV | 3.48 [3.30,3.65] | 1.43 | 2.08 | −1.46 | .95*** | .92 | .92 | .76 | .22 | .34 | .44 | .15 | .47 | .63 | .39 | .50 | (.89) |

*Note.* CI—confidence interval; LL—lower limit; UL—upper limit; General—overall work role performance.

Diagonals show AVE for scales. All correlations were significant at p <.05.

*p <.05. ***p <.001.

## Descriptive statistics and psychometric evaluation

Table 3 summarizes the descriptive statistics for the supervisory evaluations across the subdimensions of the WRPQ. Notably, ITP obtained the highest ratings among all categories. However, the differences between the subdimension results were less pronounced than those in the prior study. The distribution of all variables deviated from a normal distribution, as evidenced by the significant results of the Shapiro–Wilk test. However, despite these deviations, kurtosis and skewness demonstrated considerable improvement compared with those of the pilot study, which aligns with widely accepted standards [47].

Remarkably, the internal consistency coefficients (i.e., "Cronbach's alpha and "McDonald's omega) yielded exceptionally robust values, which confirms the reliability of the questionnaire for the individual subdimensions and the general score. These reliability coefficients, along with AVE (once again, the values are higher than correlations with other subdimensions), collectively satisfy the pre-established assumptions, which validate the psychometric soundness of the questionnaire.

Table 4 lists the findings from Study 2. The results mainly mirror the patterns observed in the pilot study. All subdimensions, except for ITP and ITA, displayed comparable mean scores and satisfactory levels of kurtosis and skewness. Although the distributions for ITP and ITA exhibited leptokurtic characteristics, which are marked by heavier tails and a more pronounced peak, the kurtosis and skewness values for the aggregate score remained within acceptable ranges [47]. Moreover, "Cronbach's alpha and "McDonald's omega exceeded the predetermined threshold. Collectively, these analyses validate the basic psychometric properties of the questionnaire, which provides the study with the confidence to proceed with further examination.

## Structure of the WRP results

To test Hypothesis 1, we conducted CFA with the *lavaan* package [45] on six potential structures for the WRPQ. The analysis employed the Weighted Least Squares Mean and Variance Adjusted (WLSMV) method, a robust estimator well-suited for Likert-scale data. These analyses were performed separately for Studies 1 and 2; Table 5 outlines the results. While all tested models exhibited statistical significance, the nine-factor solution, in which each factor corresponds to a distinct work role subdimension, demonstrated superior model fit across virtually all fit and error metrics. Notably, certain indices,

**Table 3. Scale-level descriptive statistics, study 1.**

| Scale | | Descriptive statistics | | | | | Reliability | | Correlation coefficients | | | | | | | | | |
|---|---|---|---|---|---|---|---|---|---|---|---|---|---|---|---|---|---|---|
| | | M [95% CI LL, UL] | SD | Kurt | Skew | W | α | ω | 1 | 2 | 3 | 4 | 5 | 6 | 7 | 8 | 9 | 10 |
| 1 | General | 4.1 [4.04,4.16] | 1.12 | −.67 | −.39 | .97*** | .98 | .98 | − | | | | | | | | | |
| 2 | ITP | 4.59 [4.53,4.65] | 1.12 | −.14 | −.70 | .93*** | .90 | .90 | .80 | (.87) | | | | | | | | |
| 3 | ITA | 4.34 [4.27,4.4] | 1.18 | −.27 | −.61 | .95*** | .90 | .90 | .89 | .84 | (.87) | | | | | | | |
| 4 | ITV | 3.98 [3.9,4.05] | 1.32 | −.59 | −.47 | .96*** | .91 | .92 | .92 | .68 | .81 | (.88) | | | | | | |
| 5 | TMP | 4.37 [4.3,4.43] | 1.22 | −.33 | −.64 | .94*** | .88 | .88 | .85 | .76 | .78 | .74 | (.85) | | | | | |
| 6 | TMA | 4.14 [4.08,4.21] | 1.27 | −.38 | −.57 | .95*** | .91 | .91 | .93 | .71 | .84 | .85 | .79 | (.88) | | | | |
| 7 | TMV | 3.74 [3.68,3.81] | 1.25 | −.79 | −.14 | .97*** | .88 | .91 | .88 | .58 | .71 | .85 | .68 | .79 | (.86) | | | |
| 8 | OMP | 4.08 [4.01,4.15] | 1.35 | −.55 | −.53 | .95*** | .89 | .89 | .88 | .65 | .73 | .77 | .71 | .79 | .75 | (.85) | | |
| 9 | OMA | 4.00 [3.93,4.07] | 1.26 | −.23 | −.58 | .95*** | .90 | .91 | .88 | .63 | .74 | .78 | .7 | .83 | .74 | .78 | (.87) | |
| 10 | OMV | 3.64 [3.56,3.72] | 1.48 | −.95 | −.29 | .95*** | .95 | .95 | .87 | .54 | .67 | .81 | .65 | .77 | .84 | .77 | .78 | (.93) |

*Note. CI*—confidence interval; *LL*—lower limit; *UL*—upper limit; General—overall work role performance.

Diagonals show *AVE* for scales. All correlations were significant at *p* <.05.

\*\*\**p* <.001.

**Table 4. Scale-level descriptive statistics, study 2.**

| Scale | | Descriptive statistics | | | | | Reliability | |
|---|---|---|---|---|---|---|---|---|
| | | M [95% CI LL, UL] | SD | Kurt | Skew | W | α | ω |
| 1 | General | 4.42 [4.34,4.49] | .87 | .01 | −.45 | .98*** | .95 | .95 |
| 2 | ITP | 5.50 [5.44,5.55] | .62 | 7.22 | −1.93 | .77*** | .78 | .78 |
| 3 | ITA | 4.96 [4.88,5.04] | .93 | 3.49 | −1.45 | .87*** | .77 | .77 |
| 4 | ITV | 4.30 [4.19,4.40] | 1.21 | .07 | −.68 | .94*** | .84 | .88 |
| 5 | TMP | 4.97 [4.90,5.04] | .84 | 1.70 | −1.01 | .92*** | .67 | .67 |
| 6 | TMA | 4.51 [4.41,4.61] | 1.17 | 1.21 | −1.04 | .91*** | .85 | .85 |
| 7 | TMV | 3.78 [3.66,3.90] | 1.43 | −.56 | −.43 | .94*** | .95 | .95 |
| 8 | OMP | 4.20 [4.09,4.31] | 1.28 | −.19 | −.67 | .94*** | .86 | .87 |
| 9 | OMA | 4.08 [3.97,4.2] | 1.30 | .10 | −.75 | .93*** | .86 | .86 |
| 10 | OMV | 3.45 [3.33,3.58] | 1.51 | −.93 | −.25 | .94*** | .94 | .94 |
| 11 | Con | 5.51 [5.42,5.6] | 1.04 | .02 | −.64 | .96*** | .79 | .79 |
| 12 | Clarity | 5.94 [5.85,6.03] | 1.09 | .56 | −1.00 | .87*** | .87 | .88 |
| 13 | Efficiacy | 4.53 [4.45,4.61] | .96 | −.38 | −.44 | .96*** | .93 | .93 |
| 14 | Openness | 5.19 [5.1,5.28] | 1.07 | .22 | −.44 | .97*** | .81 | .80 |
| 15 | Proactivity | 4.88 [4.8,4.97] | 1.02 | .16 | −.16 | .99*** | .88 | .88 |
| 16 | Work | 4.58 [4.49,4.67] | 1.07 | .33 | −.20 | .99*** | .74 | .74 |
| 17 | Team | 4.47 [4.38,4.55] | 1.03 | .72 | .20 | .97*** | .77 | .78 |
| 18 | Organization | 4.3 [4.2,4.41] | 1.23 | .01 | −.13 | .99*** | .88 | .89 |
| 19 | Goals | 4.52 [4.4,4.65] | 1.45 | −.27 | −.52 | .93*** | − | − |
| 20 | CWB | 1.80 [1.67,1.93] | 1.54 | .64 | .82 | .90*** | − | − |

*Note. CI*—confidence interval; *LL*—lower limit; *UL*—upper limit; General—overall work role performance; Con—conscientiousness; Clarity—role clarity; Efficiacy—occupational self-efficacy; Openess—openness to changes; Proactivity—trait proactivity; Work—work organization; Team—perceived team support; Organization—perceived organizational support; Goals—goal attainment.

\*\*\**p* <.001.

**Table 5. Comparison of alternative factor structure models in two samples.**

| Model | | Study 1 | | | | | | | | Study 2 | | | | | | |
|---|---|---|---|---|---|---|---|---|---|---|---|---|---|---|---|---|
| | | df | $\chi2$ | GFI | AGFI | TLI | PGFI | RMSEA | SRMR | df | $\chi2$ | GFI | AGFI | TLI | PGFI | RMSEA | SRMR |
| single factor | | 324 | 2211.84*** | .95 | .94 | .93 | .85 | .07 | .07 | 324 | 2118.41*** | .96 | .95 | .94 | .83 | .10 | .1 |
| two factors (in-role, extra-role) | | 323 | 1978.88*** | .95 | .94 | .93 | .85 | .06 | .06 | 323 | 1947.10*** | .96 | .95 | .95 | .83 | .10 | .10 |
| three factors (levels) | | 321 | 1782.99*** | .96 | .95 | .94 | .84 | .06 | .06 | 321 | 1747.52*** | .96 | .96 | .96 | .82 | .09 | .09 |
| three factors (role types) | | 321 | 1699.57*** | .96 | .95 | .94 | .84 | .06 | .06 | 321 | 1633.54*** | .97 | .97 | .96 | .82 | .09 | .09 |
| six factors[a] (bifactorial: levels & roles) | | – | – | – | – | – | – | – | – | – | – | – | – | – | – | – | – |
| nine factors (level × role) | | 261 | 522.58*** | .99 | .98 | .98 | .69 | .03 | .03 | 261 | 442.95*** | .99 | .99 | .98 | .68 | .04 | .05 |
| nine factors (multi-group) | female | 261 | 237.46*** | .98 | .97 | .95 | .68 | .03 | .04 | 261 | 216.63*** | .99 | .98 | .97 | .68 | .05 | .05 |
| | male | 261 | 239.98*** | .98 | .96 | .94 | .9 | .04 | .04 | 261 | 219.15*** | .99 | .98 | .97 | .67 | .05 | .06 |

[a]model did not converge. ***$p$ <.001.

such as the AGFI or TLI, yielded satisfactory values only within the context of this nine-factor, theory-driven structure (>95; [48]). In summary, the findings lend robust support to Hypothesis 1.

Upon establishing the optimal nine-factor structure, the study conducted multi-group CFA to assess measurement invariance within the results of the WRPQ. Data were partitioned based on gender. The last two rows in Table 5 summarize this part of the analysis. The same fit indices utilized in the previous CFA were examined. The models demonstrated nearly identical fit indices when analyzing the results of female and male employees, which implies strong measurement invariance across groups.

### Differential work roles predictors

To evaluate Hypothesis 2, the study performed path analysis in which all subdimensions were regressed onto a set of predictors. Consistent with the original study [1], the current study assumed that the subdimensions and predictors of work roles would exhibit internal correlations among themselves. Due to the deviation from the normal distribution of studied variables, we employed the diagonally weighted least squares (DWLS) estimator, suggested for non-normal data [49; 50]. Table 6 demonstrates the findings. The predictors accounted for a relatively uniform proportion of variance across the subdimensions, in which $R^2$ ranges from.20 to.28.

Conscientiousness and role clarity exhibited notably stronger associations with ITP than did the other subdimensions ($b$ =.14, $p$ <.001; $b$ =.24, $p$ <.001), which supports Hypothesis 2a. To statistically evaluate this hypothesis, we conducted an asymptotic $z$-test comparing correlated correlation coefficients that share a common dependent variable, using Fisher's $z$-transformation and the procedure proposed by Meng et al. [51]. Both conscientiousness and role clarity showed stronger associations with ITP than the most comparable alternative predictors ($z$ = 7.09, $p$ <.001; $z$ = 2.35, $p$ <.05). However, Hypothesis 2b did not garner empirical support; work-related self-efficacy did not exhibit the highest correlation with proficiency-related roles as expected ($b$ =.22, $p$ <.001; $b$ =.13, $p$ <.01; $b$ =.12, $p$ <.05). Contrary to assumptions, adaptivity demonstrated the strongest connection with self-efficacy ($b$ =.28, $p$ < 0.001; $b$ =.21, $p$ <.001; $b$ =.24, $p$ <.001). Similarly, the results did not support Hypothesis 2c. Openness was not primarily associated with adaptivity roles; in fact, it displayed no association ($b$ =.03, ns; $b$ = −.06, ns; $b$ = −.05, ns). Unexpectedly, it manifested certain degrees of correlation with ITP and TMP roles. Consistent with Hypothesis 2d, proactivity displayed the strongest correlation with roles that inherently require proactive behavior, such as ITV, TMV, and OMV ($b$ =.28, $p$ <.001; $b$ =.26, $p$ <.001; $b$ =.24, $p$ <.001).

**Table 6. Predictors of the work role performance, study 2.**

| Variable | Individual | | | Team member | | | Organization member | | |
|---|---|---|---|---|---|---|---|---|---|
| | proficiency | adaptivity | proactivity | proficiency | adaptivity | proactivity | proficiency | adaptivity | proactivity |
| Con | **.14***** | .08* | .09* | .01 | .08 | .10* | .07 | .03 | .09* |
| Clarity | **.24***** | .11** | −.04 | .08 | −.06 | −.09* | −.02 | −.04 | −.15*** |
| Efficiacy | **.22***** | .28*** | .14** | **.13**** | .21*** | .22*** | **.12*** | .24*** | .12* |
| Openness | .09* | **.03** | .03 | .14*** | −**.06** | .01 | .01 | −**.05** | .07 |
| Proactivity | .04 | .09* | **.28***** | .06 | .25*** | **.26***** | .15*** | .22*** | **.24***** |
| Work | **.11*** | −.02 | −**.11*** | .04 | .01 | −.15*** | −.03 | .02 | −.16*** |
| Team | .01 | .07 | .05 | **.31** | **.15***** | **.09*** | .04 | .07 | .05 |
| Organization | −.11* | .01 | .10* | −.1* | .04 | .18*** | **.34***** | **.11*** | **.27***** |
| *R²* | .26 | .23 | .21 | .23 | .21 | .28 | .24 | .2 | .24 |

*Note.* Con—conscientiousness, Clarity—role clarity, Efficiacy—occupational self-efficacy, Openness—openness to changes, Proactivity—trait proactivity, Work—work organization, Team—perceived team support, Organization—perceived organizational support. Coefficients related to hypotheses are bolded.

*p <.05. **p <.01. ***p <.001.

These associations were each significantly stronger than the correlation between proactivity and proficiency at the individual level ($z = 5.19$, $p <.001$), and remained significantly stronger than proficiency at the team and organization levels ($z = 4.54$, $p <.001$; $z = 2.30$, $p <.05$, respectively). Subsequently, the study conducted an analysis of level-related subdimensions. The results only partially supported Hypothesis 2e. Although work organization exhibited the significantly strongest relationship with ITP compared with other proficiency-related roles ($b =.11$, $p <.05$; $z = 2.94$, $p <.01$; $z = 4.28$, $p <.001$, for ITA and ITV, respectively), associations with other roles were deemed to be more complex. Once again, an unanticipated issue emerged: no relationship was observed with adaptivity at the individual level ($b = −.02$, *ns*). Finally, perceived team support and perceived organizational support were mainly strongly correlated with the respective team- ($z = 2.22$, p =.027; $z = 2.94$, p =.003; $z = 1.27$, p <.05, fort proficiency, adaptivity, and proactivity respectively) and organization-level roles ($z = 9.46$, $p <.001$; $z = 1.99$, $p <.05$; $z = 3.13$, $p <.01$, for proficiency, adaptivity, and proactivity, respectively), with each association being stronger than the most comparable alternative predictor, which supported Hypothesis 2f and 2g. In summary, the majority of the discussed specific hypotheses received empirical validation. Exceptions primarily pertained to unanticipated findings related to adaptivity-type roles.

## Discussion

This study aimed to adapt the WRPQ to Polish culture by validating and confirming the adequacy of the adopted version compared with its original counterpart. To accomplish the objectives, we examined two hypotheses: the first refers to the structural integrity of the adapted WRPQ results, and the second denotes the differential prediction of work roles through various psychological and organizational constructs. The pilot and two main studies mainly confirmed the hypotheses and revealed that the performance construct in work role theory is transferable across cultures, which is an unexplored notion until now. As a result, we introduced a reliable, valid, and theory-grounded instrument for assessing work performance.

### Structure of work role performance

The results demonstrate that the WRPQ replicated the nine-factor structure of the original measure with three role types (i.e., proficiency, adaptivity, and proactivity) on three levels (i.e., individual, team, and organizational). Significantly, alternative structures, particularly the two-factor model with in- and extra-role performance, exhibited a substantially poorer fit to empirical data and frequently failed to meet the widely accepted criteria. These findings hold critical implications for the

WRPQ and the broad field of performance studies. Previous research that used the WRPQ has adopted diverse method-ologies for calculating scores for performance and work role [13,20,15,17]. However, the factor models corresponding to these approaches did not demonstrate an adequate fit to the data. This aspect questions the validity of conclusions drawn from studies that did not utilize the comprehensive nine-role model of work role performance.

Crucially, a prevailing approach in the field of performance study and human resource management is to categorize performance into two broad domains, namely, in- and extra-role performance or task and contextual performance, accord-ing to the theoretical foundation of the study [52,7,53]. However, the current findings solidly confirm that a two-factor structure does not optimally represent the data. Indeed, past studies have questioned a mere two-factor structure and its adequacy in capturing the complexity of the performance domain [54]. Specifically, previous discussions have highlighted that the broad umbrella of extra-role is insufficiently precise for a full spectrum of work behaviors, as evident in the varied and, occasionally, conflicting definitions present in the literature [55,56]. Given the compelling validation of the comprehen-sive work role theory of Griffin et al. [1], the notion that a nuanced interplay between manifestation levels and role types could address these issues is plausible. As such, the nine-role framework delineated in this theoretical model should be seriously contemplated as a foundational alternative for performance research.

## Predictors of work role performance

To examine the differential validity of the adapted measure, the study examined the relationships of the nine performance subdimensions with different predictors, which supported the majority of the hypotheses. Predictors, such as consci-entiousness, work role clarity, dispositional proactivity, and perceived team and organizational support, demonstrated stronger associations with the work roles outlined in corresponding hypotheses. These findings hold significance, as they bolster the argument for the cultural transferability of the constructs of work role performance. As such, the study repli-cated not only the structure of the questionnaire but also its associative network.

Simultaneously, we did not find support for a few of the hypotheses. Work-related self-efficacy exhibited a stronger correlation with roles associated with adaptivity than those with proficiency. This finding may reflect cultural differences in how self-efficacy is formed and expressed. In more uncertainty-avoidant contexts such as Poland, self-efficacy may depend less on individual mastery experiences and more on social persuasion or collective encouragement, which can foster confidence in navigating changing or ambiguous situations rather than performing routine tasks [57]. Moreover, high uncertainty avoidance has been shown to shift emphasis toward adaptability and flexibility as critical competencies, which could explain why self-efficacy in this context aligned more strongly with adaptive roles [58]. Next, contrary to expectations, openness to change did not substantially correlate with the subdimensions of adaptivity. One possible expla-nation is that while the Big Five structure is cross-culturally robust, the expression of traits such as Openness may differ by context. In Poland, Openness may manifest more as intellectual curiosity and a preference for refinement of existing procedures, which are more strongly connected to proficiency-related tasks than to adaptive behaviors [59]. Moreover, in societies characterized by high uncertainty avoidance, such as Poland, change tends to be approached with caution, making openness more likely to channel into careful improvement of established routines rather than proactive adaptation to new demands [58]. Finally, work organization was unexpectedly related to proactivity. One possible explanation is that in structured work environments, such as those more typical in Poland, clearly defined procedures may create conditions in which initiative is expressed within existing frameworks. As a result, work organization may foster proactive behaviors aimed at improving or extending structured tasks [60]. Moreover, a number of these findings aligned with the existing meta-analytic knowledge on the predictors of adaptive performance [61]. However, they present a challenge to the cultural independence of work roles related to adaptivity. In recent years, scholars have increasingly recognized the importance of adaptive performance in the workplace [62], which is a trend that was further accelerated by the pandemic-induced shifts in work life. Nevertheless, scholars are only beginning to understand the potential influence of these changes on the adaptive demands placed on employees.

### Implications for research on and practice of job performance

The Polish adaptation of the WRPQ holds substantial implications for applied settings and scholarly research. Consistent with the original version [1], the adapted tool comprehensively evaluates work performance, which encompasses three pivotal role types. This framework affords a nuanced understanding of the work-related behaviors of individuals in contrast to instruments that solely focus on proficiency or specialized skills. As a result, it facilitates a more rigorous and unbiased evaluation of individual performance, which serves as a vital resource for researchers and practitioners. Using detailed items and three levels of assessment, the study reduces the influence of implicit theories during assessment; therefore, achieving a broad perspective in self-evaluation is possible. Importantly, the adopted and theoretically grounded tool facilitates the standardization of research in international settings.

### Limitations and future research guidelines

This study has several limitations that offer avenues for future research. It used independent assessments from supervisors and self-raters, which could be biased, because self-ratings tend to be lenient [24]. Future studies on work roles may gain valuable insights by utilizing data on paired performance ratings to elucidate whether or not these ratings share similar patterns of relationships with other variables despite the leniency. In this manner, the theoretical foundation of work role theory can be strengthened. Second, the samples were skewed due to a shortage of manual workers and the omission of emerging categories such as gig workers [63]. These groups could be crucial in understanding the cultural adaptability of the work performance construct and could benefit future research that focuses on these frequently neglected populations.

### Conclusion

The findings indicate that the WRPQ, when adapted to the Polish cultural context, is a reliable and valid instrument for assessing work role performance. This notion underscores the cultural transferability of work role performance theory because the structural nuances and predictive patterns within this new cultural setting closely align with those of the original. Such outcomes significantly underscore the imperative of integrating methodologically sound considerations of diverse cultural contexts in the domain of performance studies and human resource management.

### Preregistration

This study was preregistered at https://osf.io/zkj8s.

### Author contributions

**Conceptualization:** Jaroslaw Grobelny, Olga Zwardon-Kuchciak, Mateusz Hauk.

**Data curation:** Jaroslaw Grobelny, Mateusz Paliga.

**Formal analysis:** Jaroslaw Grobelny, Mateusz Paliga, Mateusz Hauk.

**Investigation:** Jaroslaw Grobelny, Mateusz Paliga, Olga Zwardon-Kuchciak.

**Methodology:** Jaroslaw Grobelny, Mateusz Paliga, Olga Zwardon-Kuchciak, Mateusz Hauk.

**Project administration:** Jaroslaw Grobelny.

**Resources:** Jaroslaw Grobelny.

**Supervision:** Jaroslaw Grobelny.

**Validation:** Jaroslaw Grobelny, Mateusz Paliga, Olga Zwardon-Kuchciak, Mateusz Hauk.

**Writing – original draft:** Jaroslaw Grobelny, Mateusz Paliga, Olga Zwardon-Kuchciak, Mateusz Hauk.

**Writing – review & editing:** Jaroslaw Grobelny, Mateusz Paliga, Olga Zwardon-Kuchciak, Mateusz Hauk.

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
