## [Decision Letter · Decision Letter 0]

7 Jul 2025

Thank you for submitting your manuscript to PLOS ONE. After careful consideration, we feel that it has merit but does not fully meet PLOS ONE’s publication criteria as it currently stands. Therefore, we invite you to submit a revised version of the manuscript that addresses the points raised during the review process.

We look forward to receiving your revised manuscript.

Kind regards,

Academic Editor

PLOS ONE

Additional Editor Comments:

Dear Authors,

Thank you for submitting your manuscript “The Polish Adaptation of the Work Role Performance Questionnaire” to PLOS ONE. I have carefully reviewed your work alongside the reviewer’s detailed feedback.

Your manuscript makes a valuable contribution by adapting and validating the WRPQ in the Polish cultural context. The methodological rigor in the translation, sampling, and psychometric analyses is commendable, and the study addresses an important gap in cross-cultural HR research.

However, there are several major concerns that need to be addressed before the manuscript can proceed to the next stage. Please see below for detailed comments.

Major Points to Address

Theoretical and Conceptual Clarifications

The unexpected findings regarding the relationships between predictors and work roles (e.g., self-efficacy & adaptivity, openness & proficiency) deserve deeper theoretical discussion. Rather than attributing these solely to “contextual shifts,” please explore alternative cultural or methodological explanations and how these findings fit within the broader literature.

Factor Structure & Methodology

Since this is a cross-cultural adaptation, it would be more appropriate to conduct an Exploratory Factor Analysis (EFA) before Confirmatory Factor Analysis (CFA), to assess whether the factor structure holds in the new context. Please either conduct an EFA or provide a strong justification for the direct use of CFA.

The manuscript reports deviations from normality in several items/subscales but still applies Maximum Likelihood estimation. Please consider re-analyzing the data using robust estimation methods (e.g., MLR, WLSMV) or justify the current approach.

Model Fit Concerns

In Study 2, the SRMR value exceeded acceptable thresholds, suggesting possible model misspecification. Please explore this further, consider testing alternative models or conducting sensitivity analyses to clarify this issue.

Minor Points

Correct the typographical error: “PMA” on page 17 should be “OMA”.

Clarify how hypotheses about “stronger associations” are operationalized — what constitutes a stronger association statistically?

Consolidate redundant references to Griffin et al. (2007) for conciseness.

Summary

Overall, your manuscript has many strengths, including methodological rigor, a large and diverse sample, and excellent reporting. However, the conceptual, methodological, and reporting issues outlined above need to be addressed to strengthen the manuscript and ensure it meets PLOS ONE’s publication criteria.

Please provide a detailed point-by-point response to these comments in your revision. I look forward to reviewing your revised manuscript.

Best regards,

Reviewers' comments:

Reviewer's Responses to Questions

**Comments to the Author**

1. Is the manuscript technically sound, and do the data support the conclusions?

Reviewer #1: Yes

2. Has the statistical analysis been performed appropriately and rigorously?

Reviewer #1: Yes

3. Have the authors made all data underlying the findings in their manuscript fully available?

Reviewer #1: Yes

4. Is the manuscript presented in an intelligible fashion and written in standard English?

Reviewer #1: Yes

Reviewer #1: The manuscript entitled "The Polish Adaptation of the Work Role Performance Questionnaire" presents a culturally adapted and psychometrically evaluated version of the WRPQ for the Polish context. The authors have undertaken a substantial project with a rigorous translation methodology and commendably large sample sizes. Overall, the work addresses a notable gap in cross-cultural work performance measurement. However, several critical issues in conceptual framing, statistical methods, and reporting must be addressed to strengthen the manuscript.

1. Conceptual and Theoretical Issues

Mismatch Between Hypotheses and Findings:

The expected relationships between predictors (e.g., self-efficacy with proficiency roles, openness with adaptivity) were largely unsupported. The authors acknowledge these inconsistencies but do not sufficiently engage with their theoretical implications. A deeper reconsideration of the cultural interpretation of adaptivity and proficiency is warranted, rather than attributing discrepancies solely to "contextual shifts."

Premature Application of CFA Without EFA:

Given the cross-cultural adaptation context, exploratory factor analysis (EFA) should have preceded CFA. Direct application of CFA assumes structural invariance, which cannot be presumed when adapting psychological constructs across cultures.

2. Statistical and Methodological Concerns

Handling of Non-Normal Data:

Although significant deviations from normality (e.g., Shapiro–Wilk test, high kurtosis for ITP items) are reported, the authors continue using maximum likelihood (ML) estimation without adjustment. Given these deviations, robust estimation methods (e.g., MLR or WLSMV) would have been more appropriate and should be employed or at least discussed.

Model Fit Interpretation:

The SRMR index in Study 2 exceeded acceptable thresholds (>0.08), suggesting potential model misspecification. The authors’ justification for this (speculating about parsimony or low inter-factor correlations) is not empirically tested. Sensitivity analyses with alternative modeling approaches are recommended.

3. Specific Technical Errors and Minor Points

Typographical Error:

Page 17 refers incorrectly to "PMA" roles instead of "OMA" (organization member adaptivity).

Hypotheses Operationalization:

Hypotheses about "stronger associations" (e.g., H2a–H2g) are vague. The manuscript should specify clear operational criteria (e.g., statistically significant higher standardized coefficients).

Redundancy:

There are repeated references to Griffin et al. (2007) within individual sections that could be consolidated for conciseness.

4. Strengths

Translation Methodology:

The authors closely follow the ITC guidelines and ensure theoretical and linguistic equivalence.

Sample Size and Statistical Power:

Both pilot and main studies are adequately powered, with clear reporting of stopping rules and statistical assumptions.

Comprehensive Reporting:

The study provides detailed tables, reliability indices (Cronbach’s alpha, McDonald's omega), and AVE analyses, enhancing the reproducibility of results.

Recommendation:

Major Revisions Required

While the manuscript is promising and addresses an important research gap, major revisions are necessary to address conceptual inconsistencies, improve statistical methodology, and clarify reporting before it can be recommended for publication.

**Do you want your identity to be public for this peer review?** For information about this choice, including consent withdrawal, please see our Privacy Policy

Reviewer #1: No

---

## [Author Response · Author response to Decision Letter 1]

1 Sep 2025

Reviewer #1 Response:

We sincerely thank the Reviewer for their thorough and constructive evaluation of our manuscript. We greatly appreciate the recognition of the study’s contribution and methodological strengths. The Reviewer’s insightful comments have been invaluable in identifying areas for conceptual clarification, methodological refinement, and clearer reporting. We believe that addressing these points has strengthened the clarity and overall quality of the revised manuscript.

Comment #1: 1. Conceptual and Theoretical Issues

Mismatch Between Hypotheses and Findings:

The expected relationships between predictors (e.g., self-efficacy with proficiency roles, openness with adaptivity) were largely unsupported. The authors acknowledge these inconsistencies but do not sufficiently engage with their theoretical implications. A deeper reconsideration of the cultural interpretation of adaptivity and proficiency is warranted, rather than attributing discrepancies solely to "contextual shifts."

Response: We thank the Reviewer for this insightful comment. In the revised manuscript, we expanded our discussion of the unexpected findings to address their theoretical implications more thoroughly. Specifically, in the subsection “Predictors of Work Role Performance”, we provide a deeper cultural interpretation of the observed discrepancies. For self-efficacy, we highlight how, in uncertainty-avoidant contexts such as Poland, self-efficacy may align more strongly with adaptive rather than proficiency roles. For openness to change, we explain that in the Polish context this trait may manifest through intellectual curiosity and refinement of established procedures, which aligns more closely with proficiency than adaptivity. Finally, regarding work organization, we discuss how structured environments may foster proactive behaviors within existing frameworks rather than supporting individual adaptivity. These additions, we believe, provide a more grounded interpretation of the cultural differences, and we hope that they adequately address the Reviewer’s comment.

Comment #2: Premature Application of CFA Without EFA:

Given the cross-cultural adaptation context, exploratory factor analysis (EFA) should have preceded CFA. Direct application of CFA assumes structural invariance, which cannot be presumed when adapting psychological constructs across cultures.

Response: We thank the Reviewer for this important methodological comment. In line with the suggestion, we conducted an exploratory factor analysis prior to the confirmatory factor analysis to avoid presuming structural invariance. The results of the EFA, together with updated Table 1, have now been added to the Results section. The analysis confirmed sampling adequacy (KMO = .88; Bartlett’s χ²(26) = 947.05, p < .001), and all loadings exceeded the recommended threshold of .35 (Hair Jr et al., 2021). Only one item demonstrated cross-loading, but as the difference between the loadings exceeded .20 and both items belonged to the same role level, this was not deemed problematic for subsequent CFA. We hope that these additions address the Reviewer’s concern and provide stronger empirical support for the measurement structure.

Comment #3: 2. Statistical and Methodological Concerns

Handling of Non-Normal Data:

Although significant deviations from normality (e.g., Shapiro–Wilk test, high kurtosis for ITP items) are reported, the authors continue using maximum likelihood (ML) estimation without adjustment. Given these deviations, robust estimation methods (e.g., MLR or WLSMV) would have been more appropriate and should be employed or at least discussed.

Response: We sincerely thank the Reviewer for this valuable comment, which allowed us to improve the methodological rigor of our analyses. In response to this concern, we have replaced the maximum likelihood (ML) estimator with a more appropriate estimator for non-normal data throughout the manuscript.

First, in the CFA, we now use the robust weighted least squares estimator (WLSMV), as it is considered suitable for ordinal and non-normally distributed data. This change led to an adjustment in fit indices, which are now updated in Table 2. The revised text in the Results section reflects this modification accordingly.

Second, for the path analysis conducted to test Hypothesis 2, we also replaced the ML estimator with the diagonally weighted least squares (DWLS) estimator, as it is recommended for models involving non-normal continuous data (Baghdarnia et al., 2014; Mîndrilă, 2010).

We would like to note that the results of the path analysis remained unchanged after switching to the DWLS estimator. This is likely due to the relatively low skewness and kurtosis of the variables involved (Byrne, 2010; George & Mallery, 2010; Hair et al., 2010). We have added an explanation and appropriate citations in the revised manuscript to clarify this methodological choice.

References:

Baghdarnia, M., Doostian, Y., & Farrokhi, F. (2014). A simulation study of parameter estimation methods in structural equation modeling under ordinal and non-normal data. International Journal of Statistics and Probability, 3(3), 54–67.

Mîndrilă, D. (2010). Maximum likelihood (ML) and diagonally weighted least squares (DWLS) estimation procedures: A comparison of estimation bias with ordinal and multivariate non-normal data. IJDS, 5(1), 60–78.

Byrne, B. M. (2010). Structural equation modeling with Mplus: Basic concepts, applications, and programming. Routledge/Taylor & Francis Group.

George, D., & Mallery, M. (2010). SPSS for Windows Step by Step: A Simple Guide and Reference (17.0 Update, 10th Edition). Pearson.

Hair, J. F., Black, W. C., Babin, B. J., & Anderson, R. E. (2010). Multivariate Data Analysis (7th ed.). Pearson.

Comment #4: Model Fit Interpretation:

The SRMR index in Study 2 exceeded acceptable thresholds (>0.08), suggesting potential model misspecification. The authors’ justification for this (speculating about parsimony or low inter-factor correlations) is not empirically tested. Sensitivity analyses with alternative modeling approaches are recommended.

Response: We appreciate the Reviewer’s careful observation regarding the SRMR value in Study 2. In response to the previous comment on the handling of non-normal data, we revised our analytical approach and re-estimated the CFA using the WLSMV estimator. As a result of this change, the SRMR value is now within acceptable thresholds, and no longer suggests model misspecification. The revised model fit indices and residuals—including the updated SRMR—are presented in Table 2 of the manuscript. _____________________________________________________________________________

Comment #5: 3. Specific Technical Errors and Minor Points

Typographical Error:

Page 17 refers incorrectly to "PMA" roles instead of "OMA" (organization member adaptivity).

Response: We thank the Reviewer for this careful observation. The typographical error has been corrected, and the term now accurately refers to "OMA"

Comment #6: Hypotheses Operationalization:

Hypotheses about "stronger associations" (e.g., H2a–H2g) are vague. The manuscript should specify clear operational criteria (e.g., statistically significant higher standardized coefficients).

Response: We thank the Reviewer for this thoughtful and important observation. We agree that hypotheses involving the relative strength of associations should be clearly operationalized to ensure they are empirically testable.

In response, we revised the wording of Hypotheses 2a–2g to explicitly refer to statistically significant differences in the magnitude of standardized effects, providing clearer criteria for evaluating support. Correspondingly, we clarified our analytical approach in the manuscript: to test whether the associations of theoretically focal predictors (e.g., conscientiousness, role clarity, perceived team or organizational support) with the outcome variables were significantly stronger than those of other predictors, we conducted asymptotic z-tests comparing dependent standardized effects that share a common dependent variable. This method is based on Fisher’s z-transformation and the procedure proposed by Meng et al. (1992), which allows for valid comparisons between correlated effects in the same model structure.

To maintain clarity and interpretability, we applied this method systematically across hypotheses to compare the focal predictor with the most relevant alternative predictors, reflecting meaningful theoretical contrasts. All results of these comparisons—including z-values and p-values—are now reported explicitly in the results section (subsection “Differential Work Roles Predictors”). We hope this addresses the Reviewer’s concern and improves the clarity and rigor of our hypothesis testing.

Meng, X.-l., Rosenthal, R., & Rubin, D. B. (1992). Comparing correlated correlation coefficients. Psychological Bulletin, 111(1), 172–175. https://doi.org/10.1037/0033-2909.111.1.172

Comment #7: Redundancy:

There are repeated references to Griffin et al. (2007) within individual sections that could be consolidated for conciseness.

Response: We appreciate the Reviewer’s observation regarding potential redundancy in our references to Griffin et al. (2007). In response, we carefully reviewed the manuscript and revised the relevant sections to improve conciseness and clarity. Specifically, we ensured that all citations now follow the APA 7th edition guidelines: the year is included with each parenthetical citation, while in narrative citations, the year is omitted after the first mention within a single paragraph. This adjustment should enhance the readability of the text while maintaining proper attribution and consistency throughout the manuscript.

---

## [Editor Report · Decision Letter 1]

3 Sep 2025

The Polish Adaptation of the Work Role Performance Questionnaire

PONE-D-25-13771R1

Dear Dr. Jaroslaw Grobelny,

We’re pleased to inform you that your manuscript has been judged scientifically suitable for publication and will be formally accepted for publication once it meets all outstanding technical requirements.

Kind regards,

Zülbiye KAÇAY, assoc. prof.

Academic Editor

PLOS ONE

Additional Editor Comments (optional):

Thank you for your thorough and constructive revisions. The manuscript is now much stronger:

The addition of EFA and the use of robust estimators have improved the methodological rigor.

The expanded cultural interpretation provides meaningful theoretical contributions.

Hypotheses are now clearly operationalized, technical errors corrected, and redundant references streamlined.

Overall, the paper is scientifically sound and ready for publication.

As a minor note, please be aware of a few small typographical issues (e.g., “Efficiacy” → “Efficacy,” “Openess” → “Openness,” stray quotation marks). These can be corrected during the copyediting/proof stage and do not affect the acceptance decision.
---

## [Editor Report · Acceptance letter]

PONE-D-25-13771R1

PLOS ONE

Dear Dr. Grobelny,

I'm pleased to inform you that your manuscript has been deemed suitable for publication in PLOS ONE. Congratulations! Your manuscript is now being handed over to our production team.

Kind regards,

on behalf of

Professor Zülbiye KAÇAY

Academic Editor

PLOS ONE